

# Annual changes in biomass amount and feeding potential of shrubby rangelands in maquis formation

Fırat Alatürk[1], Hülya Hanoğlu Oral[2], Ahmet Gökkuş[1] and Baboo Ali[1]

[1] Department of Field Crops/Faculty of Agriculture, Çanakkale Onsekiz Mart University, Canakkale, Türkiye
[2] Department of Animal Production and Technologies/ Faculty of Applied Sciences, Muş Alpaslan University, Muş, Türkiye

## ABSTRACT

**Background**. This study evaluated the extent to which the endemic herbaceous and woody species of shrubby rangelands met the roughage needs of grazing animals throughout the year.

**Methods**. The biomass, botanical composition, and quality of hay were investigated in the shrubby rangelands in Paşaköy of the Ayvacık districts in Çanakkale over the course of a year. Plant samples were taken from the herbaceous species monthly and from the grazing parts of the shrubs in May and November.

**Results**. The total amount of biomass (hay + shrub) in the rangeland was found to be 30.448 kg/ha. Shrubs made up 18.78% of the rangeland, while the annual species comprised 54.96%, and perennial herbs covered 26.26% of the total biomass. Crude protein (CP) ratios of herbaceous species decreased continuously from March (13.58%) to September (6.73%), and then increased. A similar change was also seen in pure ash (PA) ratios. The CP ratios in the shrub species were high in spring and decreased in autumn and there was an irregular variation in PA rates. Oak had the highest PA ratio during the spring, while thuja had the highest ratio in autumn, and *Juniperus oxycedrus* during the winter months. In herbaceous species, cell wall components (NDF, ADF, and ADL) reached their highest levels in summer and decreased in spring and winter. However, in shrubs, these components varied according to the species and were generally lowest in spring and then increased in autumn and winter. Here, it was determined that year-round grazing is a suitable grazing system in the shrubby rangelands of the Mediterranean zone, and animals are able to find fresh forage in the rangelands due to the presence of shrubs. However, since the contribution of shrubs to the total forage production is low, additional roughage should be provided, except in the spring when the production and quality of hay increase. These practices may contribute to better livestock management.

# INTRODUCTION

Plants can be classified as herbaceous or woody species. Herbaceous species typically have soft tissues. Woody species have a large, hard aboveground section that is difficult for

Corresponding author
Fırat Alatürk, alaturkf@comu.edu.tr

animals to digest; these species include shrubs and tree species. However, woody species can also produce quality forage for animals because of their soft tissues such as fresh shoots and leaves. For example, *Paliurus spina-christi* has a protein content that is as high as clover in the spring (*Alatürk et al., 2014*). Rangelands are mostly composed of herbaceous species, and the vegetation cover is comprised of shrublands. Shrubby rangelands are very common in Türkiye, particularly in the Mediterranean Belt and transition zones (*Yılmaz, 1993*). Shrubs can survive in drought-affected ecosystems because of their deep-rooted structure. Therefore, the majority of vegetation found in these arid and semi-arid climatic conditions consists of drought-resistant and annual shrubs (*Gençtan, 2012*). These deep-rooted plants benefit from the water found in the lower parts of the soil under dry conditions. Herbaceous plants with relatively shallow roots can maintain their green status during dry seasons because they are less affected by a water deficiency. Generally, animals prefer shrubs over herbaceous species during these seasons because the green tissues are the most nutritious parts of the plant (*Kababya, Perevolotsky & Landau, 1997*). Thus, the shrubby rangelands provide green forage to animals for longer period than those rangelands consisting of only herbaceous species (*Gökkuş, 2016*). They are indispensable forage sources for animals, especially through the long dry summers (*Papanastasis et al., 2006*).

In Türkiye, shrublands maintain an area of 11.5 million hectares while rangelands cover 14.6 million hectares (*Gökkuş, 2019*). Most of the shrubby rangelands are located in the Mediterranean belt and transition zones (*Yılmaz, 1993*). Grazing is not allowed in specific locations; however, much of the area has the potential to be used for grazing because many shrubs remain green throughout the year and can produce sufficient and quality green forage to animals when the herbaceous species become dry (*Papanastasis et al., 2009*). In fact, some shrubs such as gorse have a higher protein content than that of herbaceous species (*Alatürk et al., 2014*). Shrubby rangelands are an important natural forage sources for livestock for many reasons, including those outlined above. Shrubs have the ability to survive in arid ecosystems such as the Mediterranean belt under hot and dry summer weather because of their deep-rooted systems. Ephemeral annual species take their place next to shrub covers to escape from drought conditions. Therefore, shrubs and other annual species establish the majority of vegetation in ecosystems dominated by arid and semi-arid climates in the world (*Gençtan, 2012*). Shrubs that take advantage of the water in the lower parts of soil with their deep roots experience less drought stress in dry seasons. In this respect, herbaceous plants rooted more superficially can maintain their green condition during dry seasons. Since green tissues are the most nutritious parts of the plant, animals tend to prefer shrubs over herbaceous species during these periods (*Kababya, Perevolotsky & Landau, 1997*). Thus, shrubby rangelands provide green forage to animals for a longer period of time as compared to those rangelands consisting of only herbaceous species (*Gökkuş, 2016*). This research was conducted in Canakkale, which forms the northern border of the Mediterranean belt.

Generally, *Quercus coccifera, Quercus infectoria, Phillyrea latifolia, Juniperus oxycedrus*, and *Paliurus spina-christi* are shrub species commonly found in the maquis formation, and *Sarcopoterium spinosum* and *Cistus creticus* represent the garig formation, depending on its

soil properties (*Altın, Gökkuş & Koç, 2011a*; *Alatürk et al., 2014*). Maquis elements are very common in the rangelands of Ayvacık where this research was conducted. In this area, a free grazing system was adapted by domestic and hybrid cattle throughout the year and the cattle have adapted to grazing the shrubs. Additional feed (typically oat hay) is provided to the animals in summer and winter periods when herbaceous species become dry. In this study, the amount of biomass produced by the vegetation in the protected area and the variation in its nutritional value were examined and its importance for farm animals and grazing management were evaluated.

In Türkiye, most of the rangelands are used publicly, however, the rangelands where the research was conducted in the Ayvacık district of Canakkale are owned privately. Shrubs are an important forage source for animals in rangelands with a typical Mediterranean vegetation cover (maquis). The rangelands of Ayvacık are grazed continuously throughout the year by local cattle as recommended for dry rangelands (*Altın, Gökkuş & Koç, 2011b*) and animals are able to consume their forage without restriction in this way. However, additional roughage is provided to the cattle to ensure their continuous production when feed is limited on the range. We sought to determine whether the needs of grazing cattle were met throughout the year by the feeds produced by the herbaceous and woody plant species on the range. The nutritional value of these endemic feeds and the contribution of supplemental feed to rangelands were also considered. The extent of the production–consumption balance was evaluated for the rangeland system.

## MATERIALS & METHODS

### Study area

The research was conducted between 2013 and 2015 in a shrubby rangeland approximately 78 km to the northwest of Paşaköy, which is 13 km away from the Ayvacık district of Canakkale Province. Paşaköy has an altitude of 280 m and its geographical location is between 39.527042 latitude and 26.325733 longitude. The vegetation cover of the rangelands is a typical maquis formation. The terrain is generally hilly and the soil is shallow with bedrock appearing in places. The summer heat brings drought, which rapidly affects the area and causes the early dryness of the herbaceous species in the rangelands (early May).

### Plant materials

Samples were taken from the herbaceous and shrub species in to determine the total plant production in the experimental area. A total of seven shrub species were selected for this study. These species were commonly found in the rangeland and included *Quercus coccifera, Phillyrea latifolia, Juniperus oxycedrus, Paliurus spina-christi, Spartium junceum, Anagyris foetida,* and *Quercus infectoria*. A brief description of the shrub species is given below.

### *Quercus coccifera L.*

An evergreen and densely branched shrub that can grow up to 2–3 m in height, or a tree that may reach 8–10 m in some circumstances (*Kamalak et al., 2015*). It is the most

distinctive plant species of the Mediterranean plant community in the maquis. This species of plant is very common in arid and calcareous areas (*Martinez-Ferri et al., 2004*) and can grow in altitudes of 1,500 m.  In Türkiye, it is widely found in the Mediterranean, Aegean, and Marmara regions and may also be found in the Black Sea region (*Plants of Turkey, 2014*). It is an important source of animal nutrition and is a woody species with agricultural importance in the shrubby regions of Greece (*Papachrıstou et al., 2003*). The fruits of *Q. coccifera* may be consumed and are favored by goats.

### Quercus infectoria Oliv.
The name of this plant species is derived from thuja bees because they lay their eggs on this plant. There is a fatty, sugary, proteinaceous layer around the eggs with an outer layer of grain surrounding it. Bees spend their larval and pupal stages inside the thuja, which they will cut and emerge from when they become adults. The thuja of this plant species is used in medicine for several purposes (*Redwane et al., 2002*). In past, the thujas, which contain very high level of tannins, were used in leather, dyeing, and medicine production. *Q. infectoria* can grow at altitudes of 100–900 m. and is widely distributed in the western Black Sea and southeastern Anatolia regions in Türkiye.

### Phillyrea latifolia L.
One of the evergreen shrub species of the maquis formation. It forms a wide crown and may grow 5–6 m tall in altitude of up to 1,400 m. Its leaves and shoots are easily grazed by animals because there are no thorny structures on the edges of its leaves. *P. latifolia* is highly resistant to drought (*Ogaya & Peñuelas, 2003*) and is generally distributed in the coastal areas of Türkiye.

### Spartium junceum L.
This plant is typically found in the coastal regions of Türkiye. It is a densely-branched maquis evergreen plant that can grow up to 3 m in height and 4 m in length. It has sharp, fragrant, bright yellow flowers; the stems are green and the leaves are small and sparse. Flowers typically bloom in spring and summer. *S. junceum* is especially common in areas close to the sea and in salty soils. This species of plant contains 15 different alkaloids, mainly cytisine; however, its alkaloid rate is lower in its fresh shoots (*Greinwald et al., 1990*). Animals do not graze on this plant species willingly because of its alkaloid content. *S. junceum* has been used as an ornamental and medicinal plant (*Cave, 1995*).

### Paliurus spina-christi Miller
This plant species is found in almost every region of Türkiye. This thorny shrub sheds its leaves in winter and can grow up to 3 m. It is used in fence construction due to its thorny structure. The fruits of *P. spina-christi* contain tannins, oil, alkaloids, and glycosides. *P. spina-christi* is an important nectar plant (*Sıralı& Deveci, 2002*) and it produces feed of sufficient quality for grazing animals. The grazing portions (fresh shoots and leaves) of *P. spina-christi* have nutrient contents of 18.2% crude protein, 4.87% crude oil, 2.059% tannins, 34.85% neutral detergent fiber (NDF), and 23.01% acid detergent fiber (ADF) in spring (*Alatürk et al., 2014*).

### *Juniperus oxycedrus L. ssp. oxycedrus L.*

This plant species can be seen almost in every part of Türkiye. It will spread in arid, open, and stony areas and can grow at altitudes up to 1,800 m. It is an evergreen, coniferous shrub or tree that can grow up to a height of 10 m. It has a slow rate of reincorporation after wild fire due to its low seed germination rate; it is resistant to cold and shade (*Mifsud, 2016*). This plant is an important food source for birds and other animals and birds are known to play an important role in the re-distribution of this plant (*McBride & Trust, 1998*). This species produces hay with a low nutrient content (*Dittberner & Olson, 1983*). Some studies have shown that this plant contains compounds that may cause abortion in cattle (*Gardner et al., 1998*). Animals rarely graze on this plant and it is consumed only in the harsh conditions of winter (*Mundinger, 1979*).

### *Anagyris foetida L.*

This plant is found in the form of small clusters and belongs to the Fabaceae family. It has a foul smell and yellowish-green color in its young stage but is green in its mature stage (*Ortega-Olivencia et al., 2005*). Its flowers are important for beekeeping due to their secretions (*Valtueña, Ortega-Olivencia & Rodríguez-Riaño, 2007*). It is typically found in the coastal areas of Türkiye. *A. foetida* is avoided by ruminant animals because of the odor produced by the cytisine alkaloid and goats are the only animal known to graze on it. Horses are the most susceptible to poisoning by *A. foetida* (*Garner, 1957*). All parts of this plant, especially the seeds, are poisonous (*Mifsud, 2016*).

## Animal materials

Cattle graze throughout the year in the region. Grazing lands are typically surrounded by shrubs or stone fences to keep the animals on privately owned rangelands in the area. The vegetation cover of the rangelands where the research was conducted was representative of the region and had an area of approximately 800 da. The rangeland was used by a total of 80 local black or crossbreed cattle of both genders and a variety of ages. Approximately 30–35 of the cattle were calf/heifer and the rest were adults. The domestic gray, local black, or hybrid cattle that were already adapted to the vegetation were grazed in the study area. The water needs of the animals were provided from small artificial ponds resulting from accumulated rain water in watering pits. Four small ponds were found near the experimental area to meet the water needs of the animals. A small stream on the rangeland was also used to provide for the water needs of the animals.

## Method

The research period spanned from March 01, 2013 to March 30, 2015. Six protected areas (plots), each 625 m$^2$ (25 × 25 m), were established to represent the vegetation found on the rangeland. These plots were used to determine the amount of biomass and the nutritional values of the hay. The study areas were surrounded by wire mesh strung 1.5 m high. Both grazing and wild animals (especially pigs) were prevented from entering the experimental area. Hay and shrub samples were taken regularly, once a month, from the beginning of the research period until the end. A total of 30 herbaceous plant samples, were collected in groups of five. Samples of hay and shrubs were taken once a month for chemical analysis

throughout the experimental period. A total of 60 herbaceous plant samples were collected using 1 m$^2$ (1 m $\times$ 1 m) frames during each sampling period, five from inside (protected area), and five from outside (grazing area) each experimental plot. The insides of the frames were cut from the bottom with pruning scissors, labeled, placed in cloth bags, and then weighed. While taking the shrub samples, the annual shoots were also cut with pruning scissors and added into the bags, along with their leaves.

## Biomass

Five randomly-selected 1 m$^2$ frames from each plot were harvested from the above soil level in order to find the yield of herbaceous species. The harvested samples were labeled and placed into cloth bags, and then the amount of green hay was recorded by weighing the samples on a rechargeable digital scale. The samples taken to the laboratory were dried in the open air and then dried again in a drying cabinet at 60 °C for 48 h and weighed (*Altın & Gökkuş, 1998*) to determine the amount of dry hay. The hay yield was calculated as kg/ha. Plant samples were taken in May and November to find the yields of the shrub species. The annual shoots from one-quarter of the branch section of the shrub species were pruned with the help of pruning scissors and placed into cloth bags. The yield of green hay was recorded by weighing samples on a rechargeable digital scale. The samples were then dried in the drying cabinet and weighed, as with the hay samplings. The hay yield per plant of the grazable parts of the shrub species were determined from these results. In order to calculate the total yield of shrub species, the numbers/da and the hay production/shrub were determined as follows:

Shrub yield (kg/ha) = number of shrubs/da × grazable amount of hay/plant.

## Botanical composition

The species composition of the harvested hay from the rangeland was found on the basis of weight. For this purpose, three of the five samples from each protected area were separated in the laboratory on the basis of their species. The samples were then dried and weighed separately in order to determine the hay yield during the blooming stage of herbaceous vegetation in April. The species composition was calculated by dividing the weight of each species to the total weight. All of the shrub species in the protected areas were counted in order to find the share of shrubs into the botanical composition. Then, the diameter of each shrub species was measured and the area they covered was determined by multiplying the total number of plants/da with the covering areas of each plant. The ratio of shrubs to vegetation cover was determined by these results.

## Crude protein (CP) ratio

The total nitrogen contents of the hay and shrub samples were determined according to the Kjeldahl method. The samples were dried, ground, and prepared for analysis and the crude protein ratio was determined by multiplying the total nitrogen by the coefficient of 6.25 (*AOAC, 1990*).

## Crude ash (CA) ratio

The plant samples obtained from the plots were dried in a drying oven for 24 h at 70 °C and were burned at 550 °C in a muffle furnace until white ash was obtained. The ash was removed and weighed after the burning process was completed. The difference between the initial and the final weight was evaluated as the total ash content (*AOAC, 1990*).

## NDF, ADF and ADL ratios

The ratios of neutral detergent fiber (NDF), acid detergent fiber (ADF), and acid detergent lignin (ADL), which constitute the cell wall components of plants, were determined according to the methods reported by *Van Soest, Robertson & Lewis (1991)*.

## Statistical analyses

A one-way ANOVA with a randomized complete block design using four replications was conducted to determine the effect of the nutrient content of the shrubs. The distribution normality was determined using the Shapiro–Wilk test. The samples were found to have a normal distribution ($P > 0.05$), and one-way ANOVA was performed. The differences were considered statistically significant at the $P < 0.05$ level. Post hoc testing was used as the number of observations was equal in all groups. The Tukey's multiple comparison test was used to determine which means differed from the rest ($P < 0.05$) (*SAS, 2011*). All analyses were conducted using the SAS statistical package program.

# RESULTS

## Biomass

There was a significant variation of dry hay yields for herbaceous species according to the year*month interaction in the shrubby rangelands where the experiment was conducted (Table 1). The dry hay yield of herbaceous species in the rangelands was 26.841 kg/ha, with the highest production levels in April. The average highest dry hay yields were in the months of March (24.885 kg/ha) and April (26.841 kg/ha), followed by February (20.179 kg/ha) and May (19.706 kg/ha), according to the monthly measurements. The lowest dry hay yields were recorded in October (13.455 kg/ha) and August (13.658 kg/ha). Changes in the dry hay yield of the rangeland according to the year showed similarity to the fresh hay yield. The dry hay yield was recorded as 16.506 kg/ha in the first year, increasing to 18.720 kg/ha in the second year. The highest dry hay yields were in March (25.158 kg/ha) and February (19.868 kg/ha) in the first year of the research when the year and month were evaluated together. The highest dry hay yields in the second year were 33.982 kg/ha and 24.612 kg/ha in April and March, respectively. The lowest dry hay yields were measured in October with 9.793 kg/ha in the first year, and in August with 10.940 kg/ha in the second year of the research (Table 2).

The difference between the amount of mass produced by the shrubs was significant according to species, month, and shrub*month interaction (Table 1). Grazing mass production was highest in the *P. latifolia* (1.049 kg/ha) and *Q. infectoria* (1.002 kg/ha) shrub species. However, the masses of *Q. coccifera* and *J. oxycedrus* shrubs produced 0.419 and 0.449 kg/ha, respectively. *Spartium junceum*, *P. spina-chiristi*, and *A. foetida* formed the

**Table 1  Variation levels of nutrient contents of herbaceous and shrub plant species.**

| Varians values/ Mean Square | Herbaceous species | *Phillyrea latifolia* | *Quercus coccifera* | *Quercus infectoria* | *Juniperus oxycedrus* | *Anagyris foetida* | *Paliurus spina-chiristi* | *Spartium junceum* |
|---|---|---|---|---|---|---|---|---|
| | | | | **Crude Protein (P-Value-Mean square)** | | | | |
| Year | 0.1040-1.3420681 | 0.9462-0.0010889 | 0.8902-0.00190014 | 0.7628-0.1250000 | 0.2853-0.1701628 | 0.3024-0.5104688 | 0.0811-1.38666865 | 0.7923-0.145278 |
| Month | 0.0001*−33.2358889 | 0.0001*−20.833872 | 0.0001*−22.082162 | 0.0001*−83.982242 | 0.0001*−8.3654648 | 0.0001*−48.311414 | 0.0001*−24.998317 | 0.0001*−24.244369 |
| Year*Month | 0.4448-0.4950714 | 0.9829-0.0706828 | 0.9996-0.0033256 | 0.9995-0.0432061 | 0.6332-0.1174356 | 0.0184-1.3591259 | 0.2875-0.5503996 | 0.9999-0.0084524 |
| | | | | **Crude ASH (P-Value-Mean square)** | | | | |
| Year | 0.9568-0.0018911 | 0.3768-0.25827372 | 0.9588-0.00006806 | 0.3811-0.17900139 | 0.9294-0.00019637 | 0.7895-0.3024 | 0.4878-0.06275059 | 0.6328-0.01609634 |
| Month | 0.0001*−15.4112123 | 0.0001*−2.0405497 | 0.0001*−3.0682186 | 0.0001*−5.2528378 | 0.0001*−1.6532451 | 0.0001*−0.0001 | 0.0001*−7.3667556 | 0.0001*−5.4134816 |
| Year*Month | 0.2367-0.8521843 | 0.9021-0.15772661 | 0.9994-0.00027715 | 0.8487-0.12873472 | 0.9994-0.00338993 | 0.9999-0.0184 | 0.9716-0.03050876 | 0.9628-0.01830192 |
| | | | | **NDF (P-Value-Mean square)** | | | | |
| Year | 0.0240-44.286598 | 0.5909-0.1763723 | 0.6254-0.5460125 | 0.5937-0.1027556 | 0.6320-0.0568355 | 0.9938-0.3906021 | 0.5969-0.4984439 | 0.4424-0.1017020 |
| Month | 0.0001*−374.222309 | 0.0001*−37.401918 | 0.0001*−28.095922 | 0.0001*−33.388709 | 0.0001*−41.679585 | 0.0001*−0.0000188 | 0.0001*−17.278592 | 0.0001*−18.321354 |
| Year*Month | 0.6614-6.310748 | 0.9815-0.1829385 | 0.9997-0.1604761 | 0.9994-0.0085859 | 0.9998-0.0172869 | 0.9995-34.9355640 | 0.9968-0.2006603 | 0.8308-0.0830322 |
| | | | | **ADF (P-Value-Mean square)** | | | | |
| Year | 0.5741-0.4884014 | 0.9650-0.0006610 | 0.7286-0.0477347 | 0.8282-0.0678347 | 0.6463-0.0664876 | 0.6424-0.7326021 | 0.2766-1.7273373 | 0.7720-0.0123738 |
| Month | 0.0001*−78.6884014 | 0.0001*−19.384421 | 0.0001*−20.840816 | 0.0001*−35.165558 | 0.0001*−11.104169 | 0.0001*−46.574764 | 0.0001*−73.868184 | 0.0001*−53.444429 |
| Year*Month | 0.8350-0.8840401 | 0.9999-0.0317112 | 0.9994-0.0546953 | 0.9997-0.0307196 | 0.9862-0.0881471 | 0.9998-0.1690926 | 0.7932-0.7668194 | 0.9497-0.0429627 |
| | | | | **ADL (P-Value-Mean square)** | | | | |
| Year | 0.0360*−1.7684536 | 0.3416-0.0647381 | 0.7091-0.0234722 | 0.6196-0.0840500 | 0.2127-0.3884730 | 0.0921-1.30680000 | 0.3420-0.2480513 | 0.7126-0.03755170 |
| Month | 0.0001*−23.0172375 | 0.0001*−11.572524 | 0.0001*−13.961689 | 0.0001*−28.588357 | 0.0001*−40.783420 | 0.0001*−6.2986941 | 0.0001*−17.076740 | 0.0001*−14.575179 |
| Year*Month | 0.0185*−0.9132172 | 0.5547-0.0625308 | 0.9992-0.0245995 | 0.9988-0.0164197 | 0.9932-0.0578800 | 0.4525-0.43024762 | 0.9724-0.0631713 | 0.9902-0.0449159 |

**Table 2  Annual variation in the amount of the biomass of herbaceous species (kgha).**

| Months | March 2013–February 2014 | March 2014–February 2015 | Mean |
|---|---|---|---|
| March | 25.158 a | 24.612 b | 24.885 A |
| April | 19.700 bc | 33.982 a | 26.841 A |
| May | 16.413 d | 22.998 bc | 19.706 B |
| June | 16.380 d | 14.543 ef | 15.462 CDE |
| July | 15.408 d | 12.765 fg | 14.087 DE |
| August | 16.375 d | 10.940 g | 13.658 E |
| September | 15.108 de | 15.067 ef | 15.088 CDE |
| October | 9.793 f | 17.117 de | 13.455 E |
| November | 11.933 ef | 17.233 de | 14.583 CDE |
| December | 15.378 d | 17.585 de | 16.482 CD |
| January | 16.560 cd | 17.310 de | 16.935 C |
| February | 19.868 b | 20.490 cd | 20.179 B |
| Mean | 16.506 B | 18.720 A | |

Notes.
Level of Significance: $P_{year}$:0.0001. $P_{month}$:0.0001. $P_{year*month}$:0.0001.
Mean square values (MS): Year: 17642,4806, Month: 23595,8061, Year*Month: 8625,2694.

**Table 3  Grazing biomass amounts of different shrub species in spring and fall (kgha).**

| Shrub Species | Spring | Fall | Total |
|---|---|---|---|
| *Phillyrea latifolia* | 1.295 a | 0.802 a | 2.097 A |
| *Quercus infectoria* | 1.187 a | 0.817 a | 2.004 A |
| *Juniperus oxycedrus* | 0.468 b | 0.430 b | 0.898 B |
| *Quercus coccifera* | 0.520 b | 0.319 bc | 0.839 B |
| *Anagyris foetida* | 0.136 c | 0.033 c | 0.169 C |
| *Paliurus spina-chiristi* | 0.088 c | 0.055 c | 0.143 C |
| *Spartium junceum* | 0.036 c | 0.018 c | 0.054 C |
| Total | 3.605 A | 2.368 B | 5.973 |

Notes.
Level of Significance: $P_{year}$:0.0001. $P_{month}$:0.0007. $P_{year*month}$:0.0082.
Mean square values (MS): Year: 186,646165, Month: 56,164869, Year*Month: 8,690755.

group with the least biomass with weights of 0.0270, 0.0713, and 0.0842 kg/ha, respectively. The total grazing mass of the shrub species was found to be higher (3.605 kg/ha) in May and lower (2.368 kg/ha) in November (Table 3).

## Botanical composition

The vegetative cover of the rangelands was found to be comprised of 81.22% herbaceous species (cereals 15.73%, legumes 12.88%, other families 52.61%) and 18.78% shrub species. A total of eight cereals, six legumes, 27 herbaceous species from other families, and 14 shrub species from 55 species were found. The most dominant species found on the range were *Avena sterilis* and *Bromus hordeaceus* from cereals, *Trifolium uniflorum* and *Vicia villosa* from legumes, *Eryngium bithynicum* and *Urtica dioica* from other families, and *Phillyrea*

*latifolia* and *Juniperus communis* from shrubs. The majority (54.96%) of the herbaceous species were composed of annual plant species (Table 1).

## Nutrient ingredients
### Crude protein (CP) ratio
There was an insignificant variation in the CP ratio of the herbaceous biomass between the year and the year*month interaction. However, there was a significant difference in terms of months (Table 1). The crude protein ratio of the plant mass was highest in early spring (March and April), decreased in summer, and increased in the fall. The CP ratio was between 13–14% in March and April, but decreased by half in September (Fig. 1). The CP ratio of shrub species varied significantly by species. However, the change in terms of year and shrub*year interaction was non-significant (Table 1). The highest average CP was found in *A. foetida* (15.59%) and *P. spina-chiristi* (14.09%) species, while *J. oxycedrus* (7.20%) and *Q. coccifera* (7.82%) had the lowest CP ratio. The changes in the CP ratios of the grazing parts of shrubs species varied by year. The CP ratio in *Q. infectoria* showed a change reflecting the shape of the letter 'V'. The CP ratio in *P. Latifolia* and *Q. coccifera* increased in April, then decreased and remained flat in summer, fall, and winter. The CP ratio in *P. spina-christi* and *S. junceum* decreased consistently from spring to the end of fall, then increased and decreased again. *J. oxycedrus* showed an increasing curve from early spring to October, followed by a decrease. However, the ratio of CP increased in May with an increase in the growth of *A. foetida*, then decreased regularly (Fig. 1).

### Crude ash (CA) ratio
The crude ash (CA) contents of rangeland hay showed significant variation by month; however, it was found to be statistically non-significant according to the interaction of year and year*month. The CA ratios varied significantly in shrubs according to their species, while the change was non-significant by year and the interaction of shrub*year (Table 1). The CA contents of rangeland hay showed a V-shaped variation, generally reaching its lowest point between August and October for both years. The CA content of the grazing portion was generally low and did not change significantly throughout the year for *P. latifolia*. *Q. infectoria* had low levels of CA during the spring with an increase in the fall. The ratio of PA of *J. oxycedrus* decreased slightly in September, but typically followed a horizontal trajectory throughout the year. The CA ratios in *Q. coccifera* were high in spring but remained low in other seasons; there was an inverse pattern seen for *A. foetida*. The CA ratios in *P. spina-christi* and *S. junceum* shrub species showed similar variations throughout the year with low levels at the beginning of their growth, and reaching their highest points in August and December (Fig. 2).

### NDF ratio
Changes in the NDF contents of herbaceous species harvested from rangelands were found to be significantly important according to year and month, but the interaction between these factors was non-significant (Table 1). The NDF ratio of herbaceous species reached a peak in August for both years. The lowest ratios of NDF were recorded in February and

**Table 4  Contribution rates of different plant species in the vegetative cover of rangelands (%).**

| Plant species | Ratio (%) | Plant species | Ratio (%) |
|---|---|---|---|
| *Cereals* | | | |
| *Avena sterilis* L.* | 4.26 | *Hordeum bulbosum* L. | 1.70 |
| *Bromus hordeaceus* L.* | 2.14 | *Dactylis glomerata* L. | 1.36 |
| *Bromus tectorum* L.* | 1.93 | *Hordeum murinum* L.* | 1.52 |
| *Bromus rubens* L.* | 1.70 | *Poa bulbosa* L.* | 1.12 |
| **Total** | | | **15.73** |
| *Legumes* | | | |
| *Trifolium uniflorum* L. | 3.21 | *Trifolium stellatum* L.* | 2.22 |
| *Vicia villosa* Roth.* | 2.66 | *Trifolium repens* L. | 1.70 |
| *Onobrychis* sp.* | 2.48 | *Lupinus albus* L.* | 0.60 |
| **Total** | | | **12.88** |
| *Other families* | | | |
| *Eryngium bithynicum* Boiss.* | 6.43 | *Tordylium apulum* L.* | 1.52 |
| *Urtica pilulifera* L.* | 3.55 | *Malva sylvestris* L. | 1.49 |
| *Centaurea cya n us* L.* | 3.45 | *Achillea millefolium* L. | 1.46 |
| *Notobasis syriaca* (L.) Cass.* | 2.95 | *Echium plantagineum* L.* | 1.36 |
| *Rumex acetosa* L. | 2.69 | *Galium aparine* L.* | 1.36 |
| *Crepis sancta* (L.) Babcock.* | 2.66 | *Senecio vulgaris* L.* | 1.28 |
| *Scandix pecten-veneris* L.* | 2.14 | *Alcea biennis* L. | 1.28 |
| *Ornith o galum nutans* L. | 2.06 | *Muscari neglectum* Guss. | 1.28 |
| *Sinapis arvensis* L.* | 2.04 | *Geranium robertianum* L.* | 1.23 |
| *Ballota acetabulosa* (L.) Bentham | 1.70 | *Euphorbia helioscopia* L.* | 1.12 |
| *Lepidium latifolium* L. | 1.67 | *Linaria pyramidata* (Lam.) Sprengel. | 1.10 |
| *Ranunculus marginatus* Da'urv.* | 1.62 | *Anemone pavonia* Lam. | 1.02 |
| *Anthemis cotula* L.* | 1.62 | *Lamium album* L. | 0.99 |
| *Plantago lanceolata* L. | 1.54 | | |
| **Total** | | | **52.61** |
| *Shrubs* | | | |
| *Phillyrea latifolia* L. | 6.12 | *Spartium junceum* L. | 0.21 |
| *Juniperus oxycedrus* L. | 3.78 | *Amygdalus* webbii | 0.20 |
| *Quercus infectoria* G. Oliver | 2.90 | *Pistacia terebinthus* L. | 0.14 |
| *Quercus coccifera* L. | 2.74 | *Pyrus elaeagnifolia* Pall. | 0.14 |
| *Capparis ovata* var. *canescens* Heywood | 0.84 | *Rosa micrantha* Sm. | 0.10 |
| *Paliurus spina-chiristi* Mill. | 0.77 | *Ruscus aculeatus* L. | 0.10 |
| *Anagyris foetida* L. | 0.66 | *Asparagus acutifolius* L. | 0.08 |
| **Total** | | | **18.78** |

*Annual plant species. 54.96% ratio of vegetative cover

March. The NDF content of rangeland hay also varied significantly according to the year. The average NDF ratio was 54.68% in the first year and 56.25% in the second year.

The NDF contents of shrubs was significant in terms of year and species, however, the interaction between them was non-significant (Table 1). *Q. coccifera* had the highest

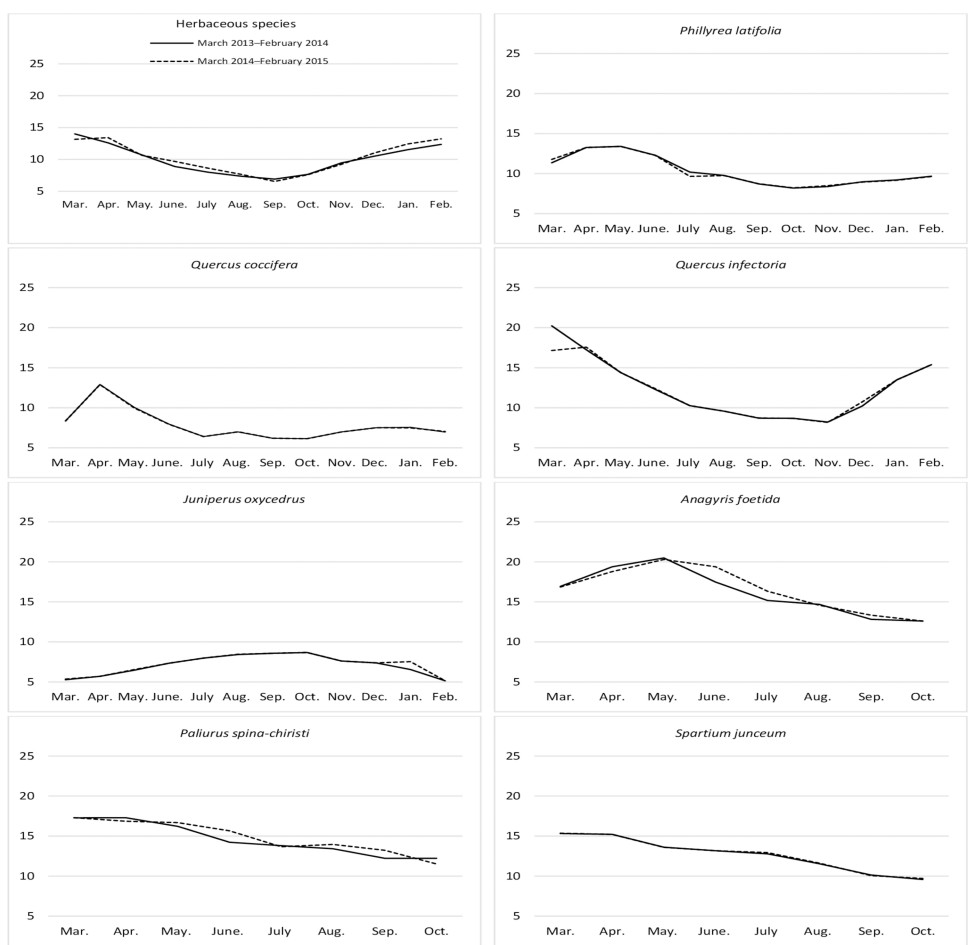

**Figure 1  Month wise variation in crude protein content of herbaceous and shrub species in range-lands.**

average NDF content (56.64%) among shrubs, and the lowest NDF ratios were determined in *P. spina-chiristi* (38.74%) and *A. foetida* (42.21%). The NDF content of shrub hay was lower in the first year (44.99%) than the second year (45.48%) (Fig. 3).

### ADF ratio

The difference between the ADF ratios of herbaceous species on the range was significant by month. The ADF ratios of rangeland hay continuously increased from March through September and October and decreased again until February. Conversely, the ADF contents of the shrub species showed non-significant changes in terms of year and year*shrub interactions, but there were significant changes based on shrub species (Table 1). Among the shrub species, *Q. coccifera* was the species with the highest ADF content (an average of 44.75%) in shrubs, while *P. latifolia* (35.60%) and *A. foetida* (32.63%) produced hay having the least ADF content (Fig. 4).

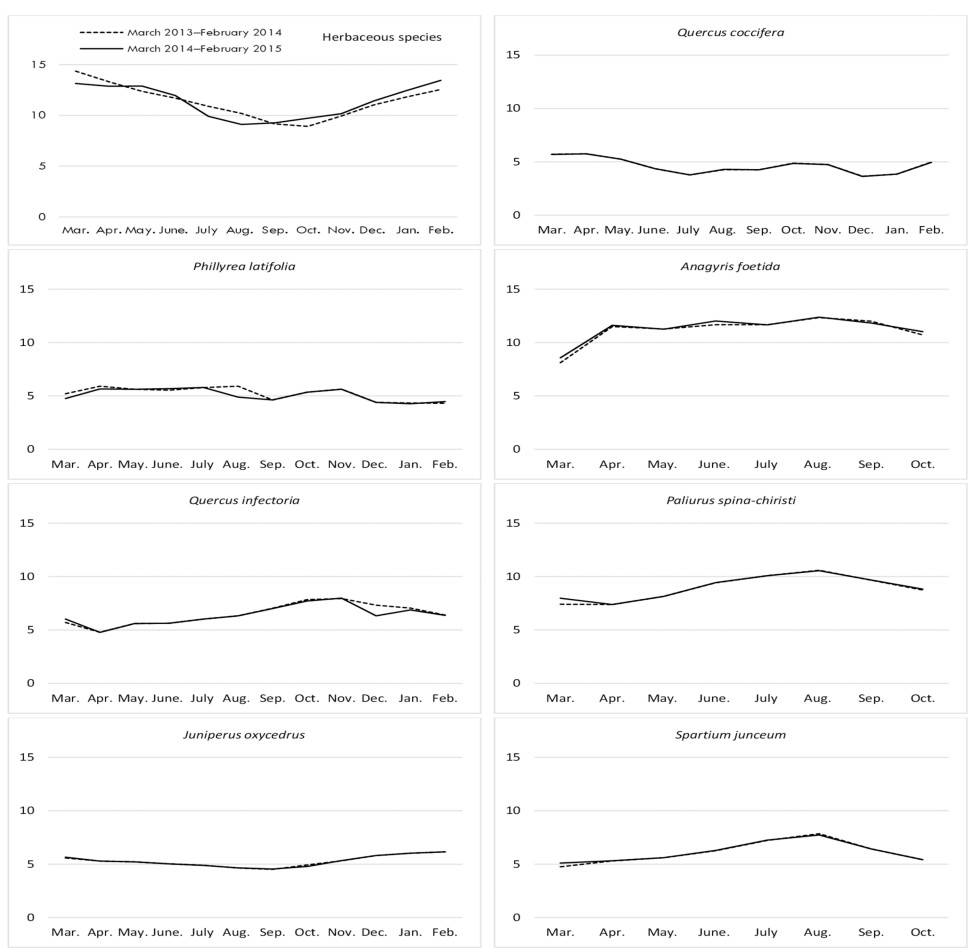

**Figure 2 Month wise variation in crude ash content of herbaceous and shrub species in rangelands.**

### ADL ratio

The ADL ratios for the year, month, year*month interaction, and the differences between them, were found to be statistically significant for the herbaceous species harvested from the rangelands (Table 4). The ADL ratios for hay increased regularly from March to November (17.81%), then decreased again until February. The hay that was fresh between the end of winter and the beginning of spring were found to contain the least lignin (Fig. 5). The ADL ratios of the different shrub species were significant in terms of year, species, and the interactions between them ($P = 0.0344$, 0.0001, and 0.0478, respectively). *J. oxycedrus* (17.40%) and *Q. coccifera* (17.16%) had the highest ADL ratios, while the lowest ADL content (11.24%) was recorded in *A. foetida*. The average ADL content (14.20%) of all shrub species was lower in first year of study than that in second year (15.27%) (Fig. 5).

## DISCUSSION

The herbaceous species had a monthly average biomass of 17.613 kg/ha over two years and the average monthly grazing biomass (total mass/12 months) of shrubs was calculated

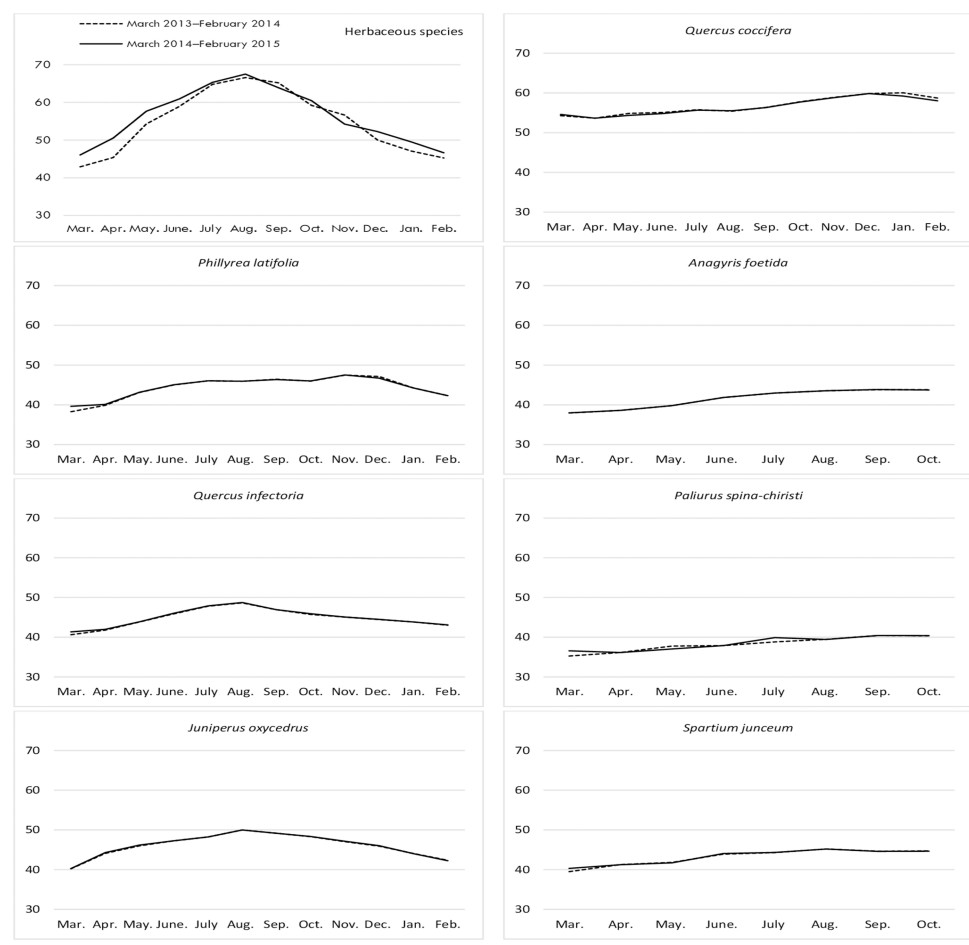

**Figure 3** Month wise variation in NDF content in herbaceous and shrub species in rangelands.

as 4.98 kg per hectare. Accordingly, the total monthly biomass of the rangelands was 18.111 kg/ha. These results show that herbaceous species constituted a large part of the hay consumed by animals, even though there was a high total mass of shrubs on the rangelands. The research area had a Mediterranean climate. Precipitation tends to fall in winter and spring and there is usually no deficiency of water in the soil during these two seasons which is why rangeland plants do not experience drought stress that will limit their growth. In addition, very low temperatures, which are the most important limiting factors for plant growth in winter, do not usually occur under Mediterranean climatic conditions. Winters are not cold in this climate, but they are generally cool. This makes it possible for cool climate plants to grow in winter, albeit slowly. For this reason, the herbaceous species continued to grow until April, following the autumn rains. The plants that make up the vegetative cover of the rangelands cannot enter into a period of rapid development until the soil temperature reaches 10 °C (*Açıkgöz, 2001*) and the soils were not sufficiently heated until the end of winter (February). During this time plants do not have adequate leaf tissue for photosynthesis, resulting in slower growth. Plants that do

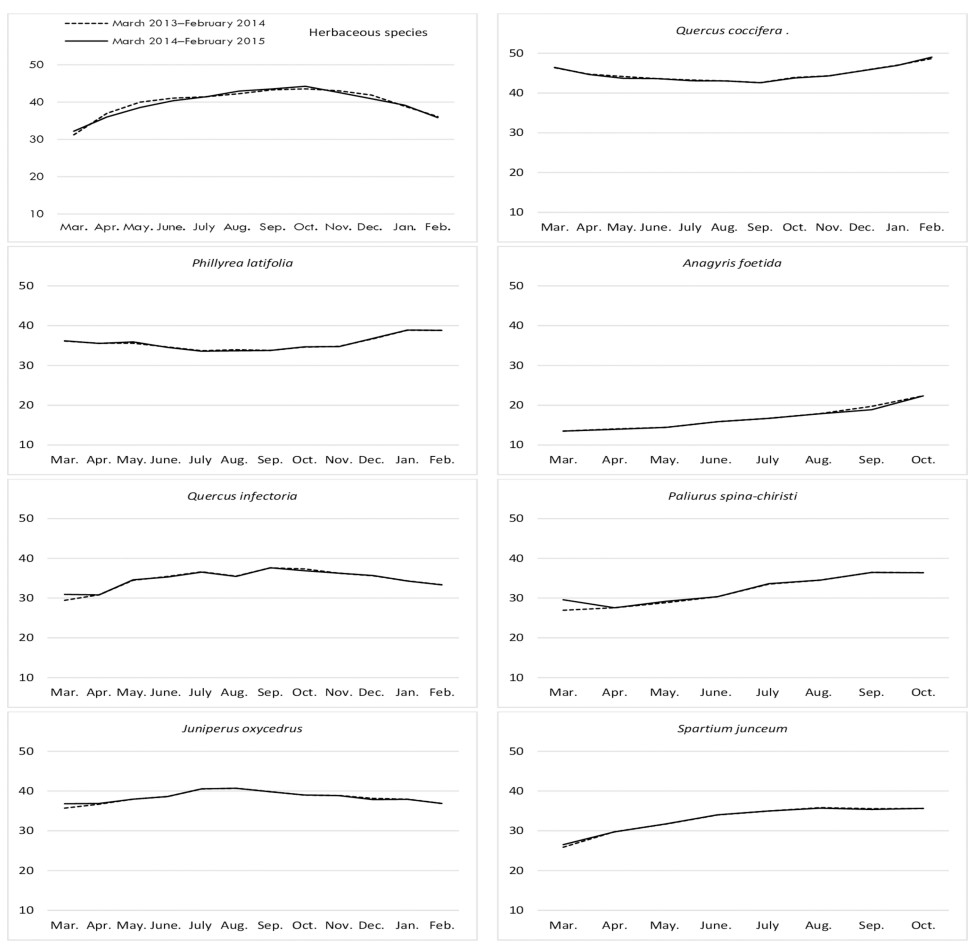

**Figure 4** Month wise variation in ADF content in herbaceous and shrub species in rangelands.

not have enough photosynthetic tissue must meet their energy requirements for growth from stored nutrients until they form these tissues (*McCarty & Price, 1942*; *Altın, Gökkuş & Koç, 2011b*; *Alatürk, 2012*). Plants grow slowly if they use reserve nutrients as a source of energy for growth, but quickly if they use photosynthesis products (*Altın, Gökkuş & Koç, 2011b*). Herbaceous plants that have sufficient moisture tend to grow during the middle of spring (in April) as a result of increased temperatures, especially soil temperatures. This rapid growth continues until the beginning of May, when the precipitation decreases and the temperature increases, at which time the plants begin to feel drought stress. This kind of growth is mostly seen in annual plant species. The herbaceous rangeland species begin to dry out along with an increase in temperatures and a decrease in precipitation. The amount of biomass also tended to decrease in May. Drought due to high temperatures is one of the most important environmental factors limiting yield in rangeland plant species (*Koç, 2001*; *Hasanuzzaman et al., 2013*; *Bhattacharya, 2019*). The negative effects of drought stress on leaf growth, photosynthesis, and the transport of assimilates are more common in rangeland plant species (*Özer, Karadoğan & Oral, 1997*). Increased stress

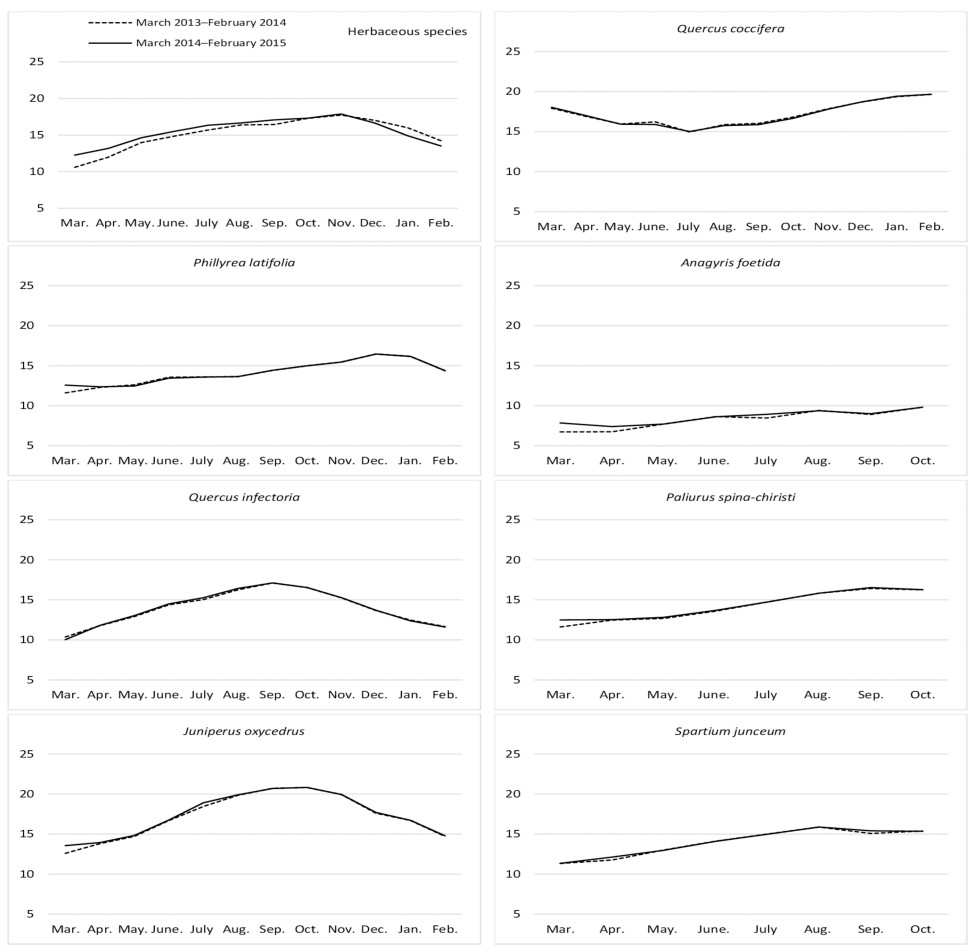

**Figure 5** Month wise variation in ADL content in herbaceous and shrub species in rangelands.

conditions enable plants (especially annuals) to rapidly transition from a vegetative to generative state (*Salisbury & Ross, 1992*). During the generative period, plants use most of the photosynthetic products for the formation of new leaves and branches, as well as flower formation, and subsequent seed filling and development (*Altın, Gökkuş & Koç, 2011a*; *Altın, Gökkuş & Koç, 2011b*; *Huijser & Schmid, 2011*). As vegetative growth slows down and eventually stops, the biomass begins to decrease. This effect was also identified in a study conducted by *Alatürk (2012)* in Çanakkale.

Air temperatures have a greater effect on shrub growth than soil moisture through the year. Shrubs in a maquis formation are true xeric species and they are less affected by drought stress as their roots extend deep into the soil (*Atalay, Efe & Soykan, 2008*; *Faber-Langendoen et al., 2016*), making the plants less dependent on precipitation as they grow. Therefore, deep rooted shrubs are among the most common types of vegetation found under arid climatic conditions (*Andiç, 1993*; *Gençtan, 2012*). Generally, shrub species start growing in April, have the greatest period of growth in May, and slow or stop growth in June, due to rising temperatures. These plants meet their water needs from the deep soil
and are able to remain green during the dry season. However, they have other nutrient needs for their growth and most of these elements are located in the upper parts of soil (*Jobbágy & Jackson, 2001*). Therefore, shrubs do not show noticeable growth, even though they remain green during summer droughts. The limited growth of herbaceous plants can be seen with autumn precipitation, which results in higher shrub yields in the spring (May) than in fall (November). However, research on lands covered with oak revealed a rate of growth of 28.9% in the region. The number of livestock grazing on this shrub was higher in September (19.10 kg/ha) than in May (13.25 kg/ha) (*Parlak et al., 2011*). Fall growth was not observed in deciduous shrubs such as Christ's thorn. The variation of the grazing biomass of shrub species is related to their genetic capacity as well as ratio in vegetation.

The maquis formation predominates the geography of the Mediterranean climatic zone. It reaches a vertical height of 700–800 m on the Mediterranean coast, 400–600 m on the Aegean coast, and 300–400 m on the Marmara coast and are spread horizontally in the inner parts where the effect of the Mediterranean climate is seen in Türkiye (*Kaya & Aladağ, 2009*). The research area was in the Mediterranean climate zone, close to the Aegean coast of the Marmara region. The hot, dry climatic conditions of summer prepare an uncompetitive environment for the deep-rooted shrubs. Therefore, approximately one-fifth (20% covered area) of the rangeland vegetation is covered with shrub species, which are maquis in nature. In arid climatic conditions annual species are prevalent and can also survive in  drought (*Fernández-Alés, Laffarga & Ortega, 1993*; *Gençtan, 2012*). Although, the total annual precipitation is sufficient, the hot, dry summers cause an increase in annual species in the geography of the Mediterranean climatic zone and the ratio of annual plant species in the Mediterranean zone constitutes approximately half or more of the total plant species in the rangelands (*Yılmaz, 1996*), which was confirmed in this study. Grazing in maquis areas increases the diversity of plant species, while the removal of animals leads to a reduction in the diversity of the plant species (*Verdú, Crespo & Galante, 2000*). The research area had a high number of animals grazed on it throughout the year.

During plant growth, when new leaf and shoot formation occurs, the nitrogen contents are higher in the newly emerged parts of the plant (*Sağlam, 2015*; *Bolat & Kara, 2017*). Therefore, the crude protein ratio is higher during the growth phase. Cell division and expansion are highest in the early growth stages (*Taiz & Zeiger, 2008*) and all of the initial biochemical reactions are catalyzed by enzymes from protein. A total of 75% of plant proteins are synthesized in the cytoplasm (*Hatfield et al., 2007*). Since the proportion of protoplasm, which forms the living part of the cells, is high at the beginning of growth, the biochemical activities and protein content are also high at that time (*Gökkuş et al., 2011*). The biochemical activities in the plant decrease and, as a result, the CP ratio decreases as the plant matures (*Angell, Miller & Haferkamp, 1990*; *Towne & Ohlenbusch, 1992*; *Yüksel & Arslan Duru, 2019*; *Grev et al., 2020*; *Chebli et al., 2022*).  Physiological activities decrease in the summer and may stop with drying. Leaves have a higher CP ratio than stems (*Mowat et al., 1965*) and as the leaf ratio of the plant decreases and the stem ratio increases in parallel with the maturation of the plants, the CP ratio also decreases  (*Buxton & Homstein, 1986*; *NRC, 2001*). Similar findings were also reported in the research conducted by *Alatürk*

*(2012)* in the same region. Variations in the CP ratios of shrubs during the year were similar to those of herbaceous species. As in the herbaceous species, the CP contents were higher in the spring months, but decreased in the summer months when the temperature was high and the precipitation was low (Fig. 1). However, these changes did not occur on a large scale as in herbaceous species because the shrubs were not as susceptible to drought the summer. In addition, the CP change in deciduous shrubs was different from that of evergreen shrubs. The CP content of the deciduous shrubs decreased continuously from the beginning to the end of their growth. Since the amount of CP in the plants is related to physiological activities, the CP ratio decreased from March to October as the physiological events slowed during the growth and maturation of the plants. *Alatürk et al. (2014)* noted a similar change in the CP contents of gorse, larch, thuja, and locust species in the same region. Young cells mark the beginning of growth in evergreen plants. During this period of growth there is an increase in protoplasm substances with a high protein content and the cell wall components are low, resulting in an increase in the CP ratio in the spring (*Papachristou, Platis & Nastis, 2005*). Depending on the progression of growth, protoplasm substances decrease and there is an increase in cell wall components (*Haddi et al., 2003*; *Parissi, Papachristou & Nastis, 2005*; *Papanastasis et al., 2008*). The highest CP ratios in spring were seen in the evergreen maple, kermes oak (*Alatürk et al., 2014*), and thuja oak (*Parlak et al., 2011*) in the Çanakkale province. The CP ratios were found to decrease continuously from the spring season in maple, menengic, kermes oak, sandalwood, and thuja oak in a study conducted in the Antalya province (*Yolcu et al., 2014*). The decrease from February–April was associated with the continued growth of plants in summer, while the CP ratios of *Juniperus oxycedrus* were higher between August and October. The CP contents of hay should not fall below 7% to allow grazing animals to adequately meet their protein needs  (*Meen, 2001*). Accordingly, herbaceous species produce protein-deficient forage in September, while shrubs are more protein-deficient in March, April, May, July, September, and February. Additional forage with higher protein levels must be given to grazing animals during these deficient periods.

The crude ash content of the herbaceous species declined with maturation, but increased as growth began in spring and fall. The need for mineral elements obtained from the soil varies depending on a plant's development during the period of rapid growth (*Alatürk, 2012*; *Gökkuş et al., 2013*; *Aydoğan et al., 2014*). The growth and development of shrubs differs from herbaceous species. However, the variation in CA ratios for shrub species was similar. The ratio of CA decreased with the increased growth in thuja, *Juniperus oxycedrus*, and gorse, while oak and maple trees had higher CA ratios in their early stages of growth (Fig. 2). These variations may be due to the geographical origin, bioclimate, and genetic structures of the species (*Laamouri et al., 2015*). Different growth patterns may also affect the root activities and ion uptake capacities. Increasing CA rates in shrubs may be due to the accumulation of mineral elements taken from the soil up to the growth tips during periods of plant growth (*Spears & Engle, 2015*) and indicate that the root activities in shrub species continue until the advanced stages of growth. Similarly, in studies conducted with oak and salty shrub species, increases in the content of mineral element may be related to maturation (*Haddi et al., 2003*; *Tolunay et al., 2009*; *Tolunay et al., 2014*). Another study conducted in

Çanakkale found that the raw ash content of oak increased with maturation (*Parlak et al., 2011*). The mineral content of plants varies according to species, developmental process, soil, and climatic conditions. A decrease in the total mineral content of *P. latifolia* in winter may be attributed to the end of the plants growth and development, resulting in stoppage of mineral uptake. Generally, the mineral content of plants varies between 8–10%. Therefore, shrub species such as *P. spina-chiristi* and *A. foetida* accumulate a sufficient amount of minerals, but shrub species such as *P. latifolia, Q. coccifera, Q. infectoria, J. Oxycedrus,* and *S. junceum* have insufficient mineral contents.

The cell wall components (NDF, ADF, and ADL) of herbaceous rangeland species increased parallel to the growth and development of the plants. The highest levels of these components were reached during drying and maturity (*Dökülgen & Temel, 2015*). The protoplasm content of cells decreases during plant growth, however, the cell walls become thickened (*Taiz & Zeiger, 2008*), indicating that the cell wall substances increase both proportionally and quantitatively. A number of studies support these findings (*Marshall, Campbell & Buchanan-Smith, 1998*; *Sarwar, Khan & Saeed, 1999*; *Kamalak, 2006*; *Mulkey, Owens & Lee, 2008*; *Parlak et al., 2011*; *Alatürk, 2012*; *Chebli et al., 2021*). The variations in the cell wall components of shrubs were similar to those of herbaceous species. However, the differences occurred due to the divergences in growth trends. In this study, NDF (cellulose, hemicellulose and lignin), ADF (cellulose and lignin), and ADL (lignin), which are the cell wall components of the grazing parts of deciduous shrubs, were at their lowest levels at the beginning of spring, but increased continuously until the end of the growth period in October, when they reached their highest points. Cell wall components, which are usually lowest in spring, increased at the end of fall and winter in evergreen shrubs. ADF and ADL contents were highest in fall and winter (Figs. 3–5).

Shrubs begin to form new shoots from March and April as the weather warmed in the spring. During this time, there is not enough ligninization in the cell walls of the shoots and the ratios of cell wall components are also lower. The weather typically begins to warm in May and precipitation decreases. This leads to the rapid maturation of herbaceous species and results in slower growth for the shrub species. There is an increase in cell wall components due to the increase in summer temperatures, decrease in precipitation, and maturation of the plant. Cells are composed mostly of water at the beginning of the growth period (*Lyons, Machen & Forbes, 1999*), and the accumulation of cell wall materials during maturation and a decrease in the leaf/stalk ratio are all indicative of these changes (*Griffin & Jung, 1983*; *Nelson & Moser, 1994*; *Açıkgöz, 2001*; *Frost et al., 2008*). Studies on shrubs grazed by animals in maquis vegetation cover typically find that the cell wall components are at their lowest levels in spring (*Huston & Pinchak, 1991*; *Steen, 1992*; *González-Andrés & Ceresuela, 1998*; *Ventura, Flores & Castanon, 1999*; *Ventura et al., 2004*; *Claessens et al., 2005*; *Pecetti et al., 2007*; *Frost et al., 2008*; *Parlak et al., 2011*; *Alatürk, 2012*; *Bouazza et al., 2012*; *Kökten et al., 2012*). The cell wall components reach their highest levels in summer or later. For instance, *Laamouri et al. (2015)* determined that the highest NDF ratios in leguminous shrubs occurred in summer and in the winter for evergreen shrub plant samples. The same results were obtained for ADF and ADL ratios (*Alatürk et al., 2014*).

## CONCLUSIONS

A total of 30.446 kg/ha of dry hay was produced in the studied rangelands and 88% consisted of herbaceous species according to the results of this study (herbaceous species in April: 26.841 kg/ha + shrub species in May: 3.605 kg/ha). The total forage value of the range consisted of herbaceous species and the grazing parts of shrubs, which is only sufficient for animals in April and May. The vegetation cover of the rangelands is mainly composed of xeric plant species. Among these, those susceptible to drought (the annual species) comprised 54.96% of the total vegetation cover, while the true xeric plant species (shrubs) made up 18.78% for a cumulative total of 73.74%. The crude protein ratios of the herbaceous species varied between 6.73–13.58% throughout the year. Among the shrub species, *J. oxycedrus* was the species with the lowest CP ratio. The crude protein ratio of hay used for animal consumption should be at least 10.60% (*NRC, 2001*). According to this parameter, the herbaceous species produced feed with CP values below the requirement for animals between June and November. *Q. coccifera* was insufficient every month of the year except April; *Q. infectoria* had insufficient levels between July and December, *P. latifolia* between July–February, *J. oxycedrus* throughout the year, and *S. junceum* in September and October. Herbaceous species contained sufficient minerals for animal consumption, with PA values varying between 9.21–13.76% by month. However, the crude ash ratios of shrubs, namely *Q. coccifera*, *P. latifolia*, *J. Oxycedrus*, and *S. Junceum*, remained at low levels throughout the year. For this reason, grazing animals could potentially experience mineral deficiencies, however, this may not be the case as the contribution of shrubs to the total hay production is lower than that of herbaceous species. However, during the summer and winter months, hay production is limited and animals may require supplementary minerals. The cell wall components of the herbaceous species were higher than the grazing portions of the shrubs, especially during their dry season. Therefore, shrubs produced forage that was more nutritious in the summer months when the herbaceous species were dry. The average quality of the hay consumed by animals should have 40–44% of NDF and 32–35% of ADF (*Putnam, Robinson & De Peters, 2008*). Hay with high NDF and ADF contents was produced in all months except March, and specifically in *Q. coccifera* throughout the year, in *Q. infectoria* and *P. latifolia* from June to December, and in *J. oxycedrus* from May to December. The herbaceous species, *S. Junceum*, had high NDF and ADF contents from August to October. Therefore, it is necessary to fill the feed deficit of animals with more quality forage in seasons other than spring when the cell wall components increase. Annual species are in the majority in shrubby rangelands with maquis vegetation cover. These species dry out quickly and lose their nutritional value as temperatures rise but comprise a majority of the grazing hay on the range and ensure a suitable grazing system throughout the year. Grazing animals benefit from herbaceous species as much as possible while they are in their fresh condition and the presence of shrubs allows animals to find fresh forage throughout the year on their rangelands. However, since the contribution of shrubs to the

total forage production is low, grazing animals should be given additional roughage except during the spring season when hay production increases.

### Funding
This article is supported by the General Directorate of Agricultural Research and Policies -TAGEM project with project number TAGEM/HAYSÜT/137 and the Carcass and Meat Quality Characteristics of Gray Breed Cattle Produced in Organic System. The funders had no role in study design, data collection and analysis, decision to publish, or preparation of the manuscript.

### Grant Disclosures
The following grant information was disclosed by the authors:
General Directorate of Agricultural Research and Policies-TAGEM: TAGEM/HAYSÜT/137.
Carcass and Meat Quality Characteristics of Gray Breed Cattle Produced in Organic System.

### Competing Interests
The authors declare there are no competing interests.

### Author Contributions
- Fırat Alatürk conceived and designed the experiments, performed the experiments, analyzed the data, prepared figures and/or tables, authored or reviewed drafts of the article, and approved the final draft.
- Hülya Hanoğlu Oral conceived and designed the experiments, performed the experiments, analyzed the data, authored or reviewed drafts of the article, and approved the final draft.
- Ahmet Gökkuş conceived and designed the experiments, performed the experiments, analyzed the data, prepared figures and/or tables, authored or reviewed drafts of the article, and approved the final draft.
- Baboo Ali analyzed the data, prepared figures and/or tables, authored or reviewed drafts of the article, and approved the final draft.

### Data Availability
  The raw data is available in the Supplementary File.

### Supplemental Information
Supplemental information for this article can be found online at http://dx.doi.org/10.7717/peerj.15204#supplemental-information.

# PeerJ

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
