# Peer review of "Annual changes in biomass amount and feeding potential of shrubby rangelands in maquis formation"

_PeerJ, doi:10.7717/peerj.15204_

## Round 0.1 · original submission · Major Revisions

Please revise it carefully according to the comments of reviewers.

Reviewer 1 ·

Basic reporting

Dear Editor,
Thanks for your invitation to revise the manuscript titled “Annual Changes in Biomass Amount and Feeding Potential of Shrubby Rangelands in Maquis Formation”. Although the manuscript provides a fairly robust dataset, it is poorly written and needs English editing.
Specific comments:
Introduction
There are several grammatical mistakes incorporated throughout the manuscript. Therefore, it needs major English editing by rephrasing long sentences, resuming paragraphs that contain less important information, and focusing on the importance of assessed issues. The introduction needs to be improved and extended. The importance of Shrubby as well as its species should be highlighted and improved. The hypothesis and rationale should be clarified. The objectives need to be revised and to be improved.
Materials and methods
This section is poorly described, needs to be carefully revised, and divided into subtitles as description of the experimental site, collected species, determined parameters, statistical analysis……
The applied experimental design should be clarified
The used unit kg/da should be converted to kg/ha
Results
The result section needs considerable improvement to be more readily
Tables should contain the MS values of the evaluated factors alongside the probabilities.
The discussion needs considerable improvement, the obtained results need to be discussed better and associated with the literature
The conclusion should be summarized and reduced
The scientific names (as (phillyrea latifolia in line 481) should be in italic throughout the manuscript

Experimental design

The applied experimental design should be clarified in Ms&Ms

Validity of the findings

OK

Reviewer 2 ·

Basic reporting

It was evaluated in the study that to what extent the herbaceous and woody species naturally found into shrubby rangelands can meet the roughage needs of grazing animals throughout the year. The manuscript both the topic selection and the research content have certain theoretical and practical significance. The data is rich, but there are many problems with the writing of the paper,
1. The research was started on March 01, 2013 and ended on March 30, 2015. In the experiment, six protected areas have been established to represent the vegetation in order to determine the amount of biomass and the nutritional values of hay. However, the time span is relatively large, because animal feeding may affect the growth of plants, and then affect the nutrition of plants, and will enclosure therefore have an impact on the nutrition of plants?
2. P87,p94:the unit of area m2.
3. p110:The note in this article uses SAS, 1999, which is recommended to be changed to the latest version.
4. P117: P=0.0001,I have my doubts as to whether the data and statistical methods are true。
5. The references are too old and lack of up-to-date references (All references are before 2015),and P45: form of reference?
6. The conclusion part of the paper is too long and complicated, and many things repeat the results. I suggest concise.
7. Figure1-5 are just trend change graphs. Can you add the results of ANOVA.

Experimental design

no comment

Validity of the findings

no comment

Additional comments

no comment

---

## Round 0.2 · Minor Revisions

The language still needs work and the reviewer still has some concerns. A minor revision is still needed.

Reviewer 1 ·

Basic reporting

The authors have addressed most of my previous concerns except certain important points. I asked the authors to add the values of mean squares (MS) of the evaluated factors alongside the probabilities in Table 1 but this has been neglected.
At the first revision, I asked to clarify the applied statistical analysis. In the revised version the authors mentioned “One-way ANOVA was carried out to determine the effect of nutrient content of shrubs”. But in Table 1 there are two studied factors and their interaction.
The manuscript still has spelling and grammatical errors and needs major English editing.
The conclusion still needs to be summarized and the discussion still needs to be improved.

Experimental design

Need to be revised

Validity of the findings

Need to be revised

---

## Round 0.3 · Major Revisions

I confirm that the authors have addressed all of the reviewers' comments. However the language quality is too poor for publication so please enlist the help of a colleague who is proficient in English and familiar with the subject matter, or contact a professional editing service to review your manuscript.

---

## Round 0.4 · accepted · Accept

I confirm that the authors have addressed all of the reviewers' comments.